# Preparation and Properties of Bio-Based Attapulgite Copolymer (BAC) Sand-Fixing Material

**DOI:** 10.3390/polym15020265

**Published:** 2023-01-04

**Authors:** Han Wang, Rui Zhao, Xiangci Wu, Dan Zhao, Hua Xue, Yuxin Zhang, Nan Dai, Dan Song, Mengling Zhang, Hui Ding

**Affiliations:** 1School of Environment Science and Engineering, Tianjin University, Weijin Road, Tianjin 300072, China; 2College of Materials Science and Engineering, Chongqing University, Chongqing 400044, China; 3Chongqing Academy of Eco-Environmental Sciences, Chongqing 401147, China; 4Huadian Aqua Membrane Separation Technology (Tianjin) Co., Ltd., Tianjin 301700, China

**Keywords:** sand-fixing materials, attapulgite, biological mycelia, 16S rDNA sequencing technology, soil feritlity

## Abstract

Desertification, one of the world’s most pressing serious environmental problems, poses a serious threat to human survival as well as to social, economic, and political development. Nevertheless, the development of environmentally friendly sand-fixing materials is still a tremendous challenge for preventing desertification. This study developed a bio-based attapulgite copolymer (BAC) by grafting copolymerization of attapulgite, starch, sulfomethyl lignin, and biological mycelia. Water retention, anti-water erosion, and anti-wind erosion tests were conducted to assess the application performance of the BAC. Scanning electron microscopy (SEM) was then employed to determine the morphology of the attapulgite and attapulgite graft copolymer sand-fixing material (CSF). The intermolecular interactions in CSF were revealed using Fourier transform infrared spectrum (FT-IR). The role of sand-fixing materials on soil physicochemical properties and seed germination was then discussed based on the germination rate experiments, and 16S rDNA sequencing technology was used to analyze the differences in microbial communities in each sample group. The results demonstrated that the BAC not only has superior application properties and significantly increased seed germination (95%), but also promotes soil development by regulating the structure of the soil microbial community. This work provides novel insights into the design of sand-fixing material for preventing desertification while improving soil fertility.

## 1. Introduction

Desertification, also known as land degradation, poses serious threats to biodiversity, food security, energy security, air quality, economic development and even the survival of humans [1,2,3]. To adhere to global sustainable development, it is therefore necessary to analyze land conditions, control land degradation, and prevent desertification [4,5,6]. Currently, sand fixation technologies can be broadly classified as: mechanical sand fixation; plant (biological) sand fixation; chemical sand fixation; and comprehensive sand fixation [7]. Mechanical sand fixation increases the roughness of the sand surface and reduces the wind speed near the ground surface by placing obstacles such as firewood, fabric, branches, and slats on the surface of the sand [8]. The technique is effective in reducing wind speed and wind erosion, as well as in controlling sand [9]. However, there are some disadvantages, such as limited sand prevention height, inconvenient transportation, excessive human and material resource consumption, etc., with mechanical sand fixation [10]. Plant sand-fixation technology is an effective method for reducing desert erosion and improving desert eco-environment quality through the planting of vegetation, which has the advantages of stabilizing quicksand, improving soil resources, extending service life, and improving the climate [11]. Due to the particular climate and soil of the desert, the survival rate of plants is low, therefore the purpose of controlling sand while only relying on plant technology cannot be achieved in practice. Chemical sand fixation involves spraying cementing materials on the sand surface to form a layer that sticks loose sand together, preventing the desertification disasters caused by wind-blown sand [12]. Chemical sand-fixing materials primarily include inorganic colloidal sand-fixing materials [13], organic colloidal sand-fixing materials [14] and organic–inorganic hybrid sand-fixing materials [15]. There are several advantages of chemical sand fixation, such as simple operation, good effects, and low cost [16]. Too many chemical additives (crosslinker and assistant crosslinker) are currently added to chemical sand-fixing materials, which threatens the survival of plants and microorganisms, so ecological restoration cannot be achieved by chemical sand-fixation alone [17].

Commonly used raw materials for sand fixation today include polymers, silicate minerals and bio-based composites. In general, polymer sand-fixing materials are macromolecular polymers, such as polyacrylamide and polyvinyl alcohol. Although polyacrylamide has enhanced sand permeability and washout resistance, chemical additions such as crosslinking agents are necessary for application, which is not in accordance with the concept of green treatment [18]. Polyvinyl alcohol not only enhances sand resistance to flushing but also significantly increases sandy soil water retention [19]; however, it is also very expensive to produce, which makes it unsuitable for large-scale applications [20]. There are active silicon oxides and aluminum oxides in silicate minerals. These materials are commonly made in two ways. One way is to prepare the material with powdered aluminum silicate as a crosslinker and calcium fluosilicate as a curing agent. The other is to add surfactants to a mixture of industrial wastes, such as slag and fly ash [21]. Sand is effectively fixed with such materials, but secondary pollution problems may arise. The most common bio-based materials for sand fixing are biological soil crusts, which are composites formed by the bonding of fungi, mosses, and their hyphae and secretions [22]. Biological soil crusts have the potential to regulate microbial communities, promote sandy soil formation, and reshape desert ecosystems by forming vegetation on desert surfaces [23]. Nevertheless, desert vegetation growth is disturbed not only by sand flow but also by the lack of water. Therefore, it is important to provide the soil crust with a water-retaining, sand-fixing environment.

According to the concept of a “combination of sand fixation and sandy soil improvement”, this paper prepared a bio-based attapulgite copolymer (BAC) by inoculating biological mycelia on sulfuric acid-modified attapulgite. This study analyzed BAC water retention, water erosion resistance, and wind erosion resistance, and compared seed germination effects of the BAC and attapulgite graft copolymer sand-fixing material (CSF), as well as changes in sandy soil bacterial community abundance and diversity after remediation with different sand fixation materials. Results of the study are expected to produce an environmental protection sand fixation material, provide new ideas for the world’s desertification control work, and provide theoretical and technical support for the restoration of desert ecosystems.

## 2. Materials and Methods

### 2.1. Materials

The attapulgite stick (APT) was procured from Anhui Lenong Environmental Protection Technology Co., Ltd. (Wuhu, China). The sulphuric acid (H_2_SO_4_, AR) and starch ((C_6_H_10_O_5_)_n_, AR) were purchased from Tianjin Jiangtian Chemical Technology Co., Ltd. (Tianjin, China). Lignin ((C_20_H_24_Na_2_O_10_S_2_, AR) was obtained from Tianjin Guangfu Fine Chemical Research Institute. (Tianjin, China). Polyacrylamide ((C_3_H_5_NO)_n_, AR) and ammonium persulfate ((NH_4_)_2_S_2_O_8_, AR) were purchased from Fuchen Chemical Reagent Co., Ltd. (Tianjin, China). The biological mycelia cultured in the laboratory was *Pleurotus ostreatus*. The sand used in the experiment came from the Tengger Desert in Gansu Province. The selected seeds were *Brassicacampestrisssp.chinensis –Makino* purchased from Tangshan Xinma Seed Industry. The experimental water was distilled water (DW).

### 2.2. Preparation of BAC

The 6 mol/L H_2_SO_4_ and APT were placed in a round bottom flask according to the mass ratio of 10:1, heated to 60 °C, stirred continuously for 60 min, and then centrifuged. The precipitate was thoroughly washed with DW to neutrality and filtered with a Buchner funnel. The solids obtained from the filtration were dried in an oven at 105 °C to constant weight. The acid modified APT were prepared as the above solids were ground and sieved with a 200 mesh sieve.

A four-necked flask was prepared with a stirring paddle, a condenser, a thermometer, and an air outlet with which to mix the starch and DW in a 1:10 mass ratio under nitrogen atmosphere. The system was continuously heated to 95 °C, stirred for 2 h, and cooled to room temperature. The 4 g APT modified with 6 mol/L sulfuric acid and 10 g lignin were added to the continuous stirring system, where their mass ratios to the DW were 4% and 10%, respectively. After being stirred for 30 min and heated to 70 °C, the 3 mL polyacrylamide (PAM) and 40 μL ammonium persulfate were slowly added and stirred continuously until they were well blended. Then, 20 μL ammonium persulfate was added to the solution described above. The pH of the solution was adjusted to neutral with 1M NaOH and put into the oven at 95 °C for 1 h. The solid in the solution was extracted with anhydrous ethanol, vacuum filtered and washed with DW, and dried in the oven at 60 °C until it reached a constant weight, which was the CSF used in the subsequent experiments.

According to the method of Zheng et al., 1 × 10^8^ CFU (colony forming unit) of *Pleurotus ostreatus* were inoculated into each kg of dry corn cob powder, then the biological sand-fixing materials (BSF) were prepared after over seven days of natural compost fermentation [24]. The 3% mass concentration of CSF was uniformly sprayed into BSF to create BAC.

### 2.3. Application Performance Experiment of BAC

#### 2.3.1. Water Retention Experiment

In the laboratory, the sand column model was prepared (Height: 35–45 mm, diameter: 90 mm, sand weight: 76 g, particle size: 30–100 mesh, petri dish weight: 44 g) (Figure 1a) for testing the water retention of the sand-fixing materials [25]. CSF and BAC were evenly sprayed over the sand column model, and the weight of the sand fixing system was recorded within 55 h (average of the three measurements), the water content of the sample was calculated as follows:ω = (M_0_ − M)/(M − m) × 100%(1)
where ω is the water content of the sand column model; M_0_ and M are the weight (g) of the sand column model (including the petri dish) before and after water evaporation, respectively; m is the weight (g) of the petri dish.

#### 2.3.2. Anti-Water Erosion Experiment

The anti-water erosion device was assembled as shown in Figure 1b in the laboratory to test the sand column’s resistance to water erosion. Before the experiments, CSF solutions with mass concentrations of 0, 1, 2, 3, and 4%, as well as BAC solutions of 3% were evenly sprayed on the sand column model and dried in the oven at 50 °C for 12 h. In the experiment, a uniform water distributor was used to simulate rainfall of 10 L/m^2^ to test the sand-fixing material’s anti-water erosion. The weight of the sand-fixing system was recorded within 25 min (the average value of three measurements), and the experiment was repeated three times.

#### 2.3.3. Anti-Wind Erosion Experiment

The anti-wind erosion device was assembled, as shown in Figure 1c, in the laboratory to test the sand column’s resistance to wind erosion. In preparation for the experiment, 1, 2, 3, 4 and 5% BAC solutions were evenly sprayed on the sand column model and dried at 50 °C for 12 h to simulate wind blowing (12, 14, 16, 18 and 20 m/s) for the anti-wind erosion experiment. The weight of the sand-fixing system was recorded within 40 min (the average value of three measurements), and the experiment was repeated three times.

#### 2.3.4. Stability Experiment

Due to the high temperature in desertification, the sand-fixing materials require own good thermal stability. The BAC was tested for thermal stability at different temperatures, which reflected the sand-fixing capability of the material: the prepared sand-fixing material was placed in a glass bottle and adjusted to 0, 10, 20, 30 and 40 °C, respectively, for 48 h, and observed for decomposition.

### 2.4. Characterization of Sand-Fixing Materials

The structure and sand-fixing ability of APT before and after acid modification were observed with the Hitachi S-4800 scanning electron microscope (SEM, Hitachi Co., Tokyo, Japan). A Fourier transform infrared spectrometer (FT-IR, Tianjin Gangdong Sci. &Tech. Co., Ltd., Tianjin, China) was used to analyze the chemical interaction between APT, starch, lignin and PAM functional groups in the CSF.

### 2.5. Effects of Sand-Fixing Materials on Sandy Soil Physicochemical Properties and Seed Germination

Four treatments were set up in the experiment: (1) no treatment, recorded as Blank; (2) sand sample treated with CSF, recorded as CSF; (3) sand sample treated with BSF, recorded as BSF; (4) sand sample treated with BAC, recorded as BAC. To begin with, the physicochemical characteristics of the four treatments were determined according to the following methods. The organic matter (OM) content was determined with the potassium dichromate volumetric method, the pH value was measured by the glass electrode method and total nitrogen (TN) was determined by the Kjeldahl method. The available phosphorus (Olsen-P) and available potassium (Avail-K) levels were determined by atomic emission spectrometry. The bulk density of the sandy soil was measured by the pit digging method, and the sandy soil porosity was calculated at the same time. Meanwhile, a 10 cm diameter petri dish was prepared, covered with test sand, moistened, and the seeds were buried 1 cm deep in the dish. Four treatments were designed in the irrigation experiment: (1) 0 times: no watering; (2) 1 time: watered 20 mL on the first day; (3) 2 times: watered 20 mL on the first and third day; (4) 3 times: watered 20 mL on the first, third and fifth day. Petri dishes are separated by an isolation distance to prevent mutual influence. Germination rates were counted every day after sowing at 20 o’clock, and the final germination rate was counted seven days later.

### 2.6. Acquisition of Samples and Misep Sequencing of the 16S rDNA Genes

Five sandy soil samples were prepared. In addition to the four sandy soil samples in the above seed germination experiment, the soil around the roots of plants in the BAC group was taken as the fifth sample. A stainless-steel ring knife (5 cm in diameter) was used to collect sandy soil samples from the surface sandy soil (0~10 cm). The samples were collected from each group of sand plots in S-shaped mode and then mixed.

According to Yu et.al, DNA extraction and PCR amplification should be conducted on the collected sample [26]. Two-thirds of the sample was placed into a 50 mL EP tube. The freshly prepared PBS buffer solution was added, shaken for 10 min and left to stand for 30 s. The upper suspension was then removed, centrifuged for 5 min, and the precipitate was taken out. The DNA was extracted according to the instructions of the E.Z.N.A.^®^ soil kit (Omega Bio-tek, Norcross, GA, USA), DNA concentration and purity were determined by NanoDrop2000, and DNA quality was determined by 1% agarose gel electrophoresis. Assay parameters: voltage 5 V/cm, time 20 min; primers 338F (5’-ACTCCTACGGGAGGCAGCAG-3’) and 806R (5’-GGACTACHVGGGTWT-CTAAT-3’) for PCR amplification of the V3-V4 hypervariable region. The amplification procedure was as follows: 3 min at 95 °C, followed by 27 cycles of 30 s at 95 °C, 30 s at 55 °C, and 45 s at 72 °C, and a final extension at 72 °C for 10 min. The PCR products were recovered on 2% agarose gel and purified using AxyPrep DNA Gel Extraction Kit (AxygenBiosciences, Union City, CA, USA). Quantitative fluorescence TM-ST (Promega) was used for detection and quantification. The sequencing was performed on the Illumina MiSeq platform of Majorbo BioPharm Technology Co., Ltd. (Shanghai, China).

## 3. Results and Discussions

### 3.1. Application Performance of BAC

According to the results of water retention experiments in CSF and BAC (Figure 2a), the sand becomes dehydrated over time. BAC-treated sand had a higher water content than CSF-treated sand throughout the experiment, and the sand treated with CSF was nearly dry after about 20 h, whereas BAC-treated sand was completely dry after 50 h. As a result, BAC showed superior water retention properties over CSF. As shown in Figure 2b, as a widely used sand-fixing material, CSF’s anti-water erosion ability increased with concentration. The water retention performance of sandy soil treated with BAC at 3% concentration is comparable to that of sandy soil treated with CSF at 2–3% concentration, indicating that BAC has good water erosion resistance. It is illustrated in Figure 2c that BAC achieved a sand fixation rate of more than 70% at wind speeds of 12, 14, 16 and 18 m/s. At a wind speed of 20 m/s, BAC still had a sand-fixing effect of 40%, demonstrating the material’s anti-wind erosion properties. Figure 2d shows that the BAC has good stability below 20 °C, delamination occurs at 30 °C, and decomposition occurs at 40 °C.

### 3.2. Analysis of Characterization Results

#### 3.2.1. SEM

In Figure 3a,b, APT modified with acid shows obvious agglomeration, sheet-like structures and rod-like crystals are reduced, and local structural fractures and collapses as well as depolymerization of mineral aggregates are visible. In the process of modifying APT by acid, the channels of the material were dredged through the decomposition of carbonate minerals and the replacement of cations. This increased the sand-fixing performances of the material, especially its ability to absorb and retain water. The SEM images of CSF with APT, starch, lignin and PAM in Figure 3c show that starch, lignin, and APT were accommodated in the network of PAM, and the surface of the material is smooth and has loose channels. Due to the lack of PAM support, the starch, lignin, and APT in Figure 3d appears lamellar. Despite this, they have large channels and rough surfaces, proving the material’s strength in absorbing water. However, the wind erosion resistance and water erosion resistance became worse with the lack of PAM support. Figure 3e shows a mosaic combination of PAM and lignin with the graft copolymer molecules as a network structure formed by cross-linking the molecular chains. As seen in Figure 3f, APT fiber clusters and many small pores appear on the originally smooth PAM surface, which indicates that the material is capable of absorbing water.

#### 3.2.2. FTIR

Figure 4 shows that the sharp band in the APT spectrum is centered at 3620 cm^−1^, and after the graft copolymerization of APT, the O-H group disappeared here. Furthermore, APT exhibits a strong absorption peak at 3420 cm^−1^, which is attributed to the stretching vibration absorption by the amide group (N = H). There are three absorption peaks in the figure at 1663 cm^−1^, 1558 cm^−1^ and 1239 cm^−1^, which correspond to amide I, amide II, and amide III, respectively [27]. In contrast to PAM-APT-lignin, the absorption peaks of PAM-APT-lignin-starch at 1684 cm^−1^, 1449 cm^−1^ and 1240 cm^−1^, as well as a blue shift of 20 cm^−1^ [28], possibly due to overlapping APT and PAM carbonyl absorption peaks, indicates that the graft copolymerization reaction between these molecules occurred.

### 3.3. Physicochemical Properties of Sandy Soil

The basic physicochemical properties of sandy soil under four treatments are shown in Table 1. Compared with the Blank group, the OM content of sandy soil added with CSF and BSF increased significantly. To eliminate the influence of raw material starch, the OM content of the BAC and CSF groups was compared. The former had a higher OM content than the latter, indicating that mycelium significantly increased the microbial and OM content in the soil. Because the culture environment of mycelium requires the pH to be 3.5~4.5, the pH of the soil was adjusted during the culture of mycelium. Therefore, the sandy soil treated by BSF was weakly acidic, while the other groups of sandy soil were neutral. Compared with the Blank and CSF group, TN, Olsen-P, and Avail-K levels in BSF and BAC group increased significantly, which indicated that BSF could increase the contents of nutrient elements in soil, which was beneficial to plants and microorganisms [29]. From the particle size composition of sand, compared with Blank, the bulk density of CSF is 1.5 g/cm^3^, the porosity is reduced, the sand particles are enlarged, and the sand is bonded together. This proves the sand-fixing effect of CSF, but also that it is too tight sand hinders plant growth [30]. When CSF and BSF are combined, sand bulk density, sand aggregate structure, and sand porosity are all improved.

### 3.4. Analysis of Experimental Results of Germination Rate

Many factors affect seeds germination, including irrigation times, treatment methods, and burial depth. Irrigation times and sand treatment methods were the variables in this experiment. As water plays a crucial role in seeds germination, the seeds of all four groups hardly germinate when there is a lack of water (0 times). In contrast, when the amount of water increased, the germination rate increased exponentially, particularly in Figure 5c,d. Among the different sandy soil treatments, the Blank group showed the lowest germination rate (20%) and the BSF and BAC group showed the highest (95%). As can be see in Figure 5a,b, the seed germination rate of the CSF group is higher, indicating the CSF can retain water, which is beneficial to seed germination. The germination rate of the CSF group (41%) was lower than that of the BAC group (95%) suggesting that mycelium had a positive effect on seed germination.

### 3.5. Analysis of Abundance and Diversity of Bacterial Community before and after Sandy Soil Treatment

#### 3.5.1. Statistics and Analysis of Sequencing Data

Sequences were randomly selected from the sample to determine the number of species, and then the rarefaction curve were constructed based on the number of species and sequences. There was a sharp rise followed by a plateau in the rarefaction curve as the number of sequences increased (Figure 6a), indicating that the sequences obtained in this experiment reflected the bacterial population structure of the sample environment. The Shannon index curve (Figure 6b) showed the microbial diversity of samples in different sequencing quantities. As the number of sequences increased, the Shannon index curve rose first and then tended to be flat, indicating that the sequencing results of this study meet the criteria for analysis.

#### 3.5.2. Operational Taxonomic Unit (OTU) Statistical Analysis

As can be seen from Figure 7a, the number of OTUs in different samples is quite different. The number of OTUs in the CSF group is similar to that in the Blank group, while the number of OTUs in the BSF group, BAC group and Plant group increased significantly compared with the Blank group. These results indicated that the species abundance of the last three groups of sandy soil significantly increased after improvement. In the Venn diagram, similarities and differences between the microbial communities observed in different samples are depicted. As shown in Figure 7b, the total number of OTUs in five samples is 52, and the unique OTUs in Blank, CSF, BSF, BAC and Plant samples were 12, 72, 32, 26 and 156, respectively, accounting for 6.1%, 33.3%, 9.7%, 6.3% and 27.7% of the total number of OTUs in each sample. The number of OTUs shared by BAC and Plant and not shared by other samples was 106, which was quite different from the microbial communities of other samples.

#### 3.5.3. Analysis of α Diversity

The Chao1 index, ACE index, and Shannon index were used to measure species richness and diversity. The Chao1 index and ACE index are directly proportional to the richness of the microbial community, while the Shannon index is directly proportional to the diversity of the microbial community. In Table 2, the coverage rate of all samples was 99%, the database was large, and the data were accurate, reflecting effectively the diversity of microflora. The Chao1 index, ACE index and Shannon index were higher in the BSF and BAC groups than the CSF group, indicating that the addition of mycelium increased the species richness and diversity in the soil. The Plant group had the highest Chao1, ACE and Shannon indices, indicating that the entry of BAC into the soil environment increased the species richness and diversity of the plant-rooted soil and promoted soil development.

#### 3.5.4. Analysis of Bacterial Community Structure Based on Phylum, Phylum and Genus

Figure 8a shows the microbial composition of the bacterial community at the phylum level. All samples contain the highest amount of *Proteobacteria*, ranging from 35.5% to 67.3%. Interestingly, *Firmicutes* have been found in 44.3% of sand samples improved by BAC. As many *Firmicutes* can produce spores resistant to dehydration and extreme environments [31], the high abundance of *Firmicutes* indicates that some microorganisms in the sandy soil treated by BAC had changed to cope with the environment. Another dominant bacterial phylum is *Bacteroidetes*, which degrades organic matter. Due to the high organic content of raw material for CSF and BAC, *Bacteroidetes* abundance increased significantly in the treated sandy soil. Further comparative analysis of the microbial community at the phylum level are shown in Figure 8b. All samples were dominated by *Actinobacteria* and *Bacteroidetes* at the phylum level. There was a significant difference in the microbial abundance in the samples before and after the sandy soil treatment, with the emergence of *Gemmatimonadetes* and *Saccharomyces* that were not present in the Blank samples. The BAC group has the highest percentage of *Bacillibacteria*, which is a beneficial soil microorganism that can effectively remove soil organic pollutants while also contributing to the restoration of the soil ecology [32]. The top 50 taxonomic genera and five samples were hierarchically clustered using the Bray–Curtis similarity index to analyze microbial community profiles by genus (Figure 8c). As a result of cluster analysis, it was determined that the microbial composition of the sandy samples treated with CSF did not resemble other groups. Representative genera in the CSF group samples were *Bosea*, *Ochrobactrum*, *Brevundimonas*, and *Sphingobacterium*, while representative genera in the Blank group were *Paracoccus*, *Cellulomonas*, and *Paenibacillus*. However, it seems that the composition of the two samples is relatively similar because 15 of the top 50 genera are shared. As the BAC-treated sand samples were most correlated with the Plant samples, this suggested that vegetation on the treated sandy soil positively contributes to soil development. There is a close relationship between sandy soil samples modified by BSF and sandy soil samples modified by BAC, since the BAC material is formed by adding biological mycelia to the CSF material.

## 4. Conclusions

In this study, biological mycelia were applied to attapulgite copolymer to prepare a low-cost and environmentally friendly sand-fixing material (BAC), which holds the promise of realizing the vision of sand fixation and soil remediation.

In the application performance experiments, the water in the BAC group evaporated completely after 50 days, and the water retention time was twice that of the CSF group. The BAC still had a sand-fixing effect of more than 70% at winds of 18 m/s. It showed good stability at temperatures of 20 °C. The BAC can increase soil fertility, provide more nutrients, and provide vegetation with a better environment. The sandy soil treated with BAC showed a significant increase in soil OM content and TN, Olsen-P and Avail-K content, as well as an improvement in soil bulk density, soil porosity and soil agglomerate structure. Germination rate experiments verified the positive effect of the addition of biological mycelia on seed germination, with 95% germination at 7 days in the BAC and BSF compared to 41% in the CSF group. High-throughput sequencing of 16S rDNA demonstrated greater soil microbial diversity after BAC treatment. Additionally, the samples of the BAC group and Plant group were similar, proving the importance of vegetation in soil development. In summary, BAC contributed to sand fixation and soil restoration. This article will provide the theoretical basis and technical guidance for turning desertification areas into fertile fields.

## Figures and Tables

**Figure 1 polymers-15-00265-f001:**
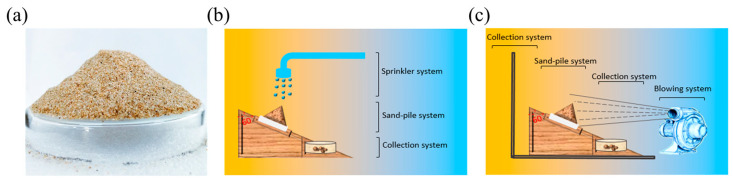
(**a**) Sand column model; (**b**) Device for testing anti-water erosion; (**c**) Device for testing anti-wind erosion.

**Figure 2 polymers-15-00265-f002:**
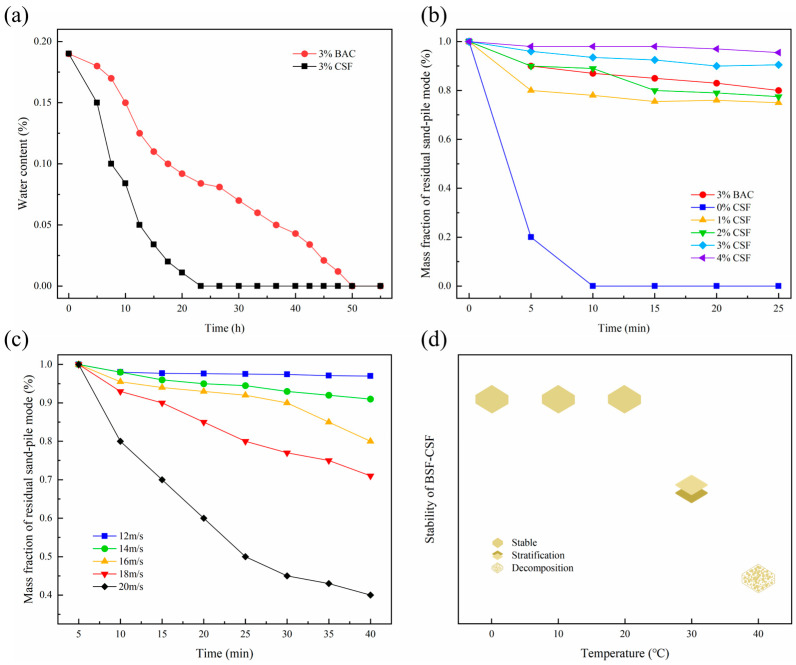
Application performance: (**a**) water retention experiment; (**b**) anti-water erosion experiment; (**c**) anti-wind erosion experiment; and (**d**) stability experiment of BAC.

**Figure 3 polymers-15-00265-f003:**
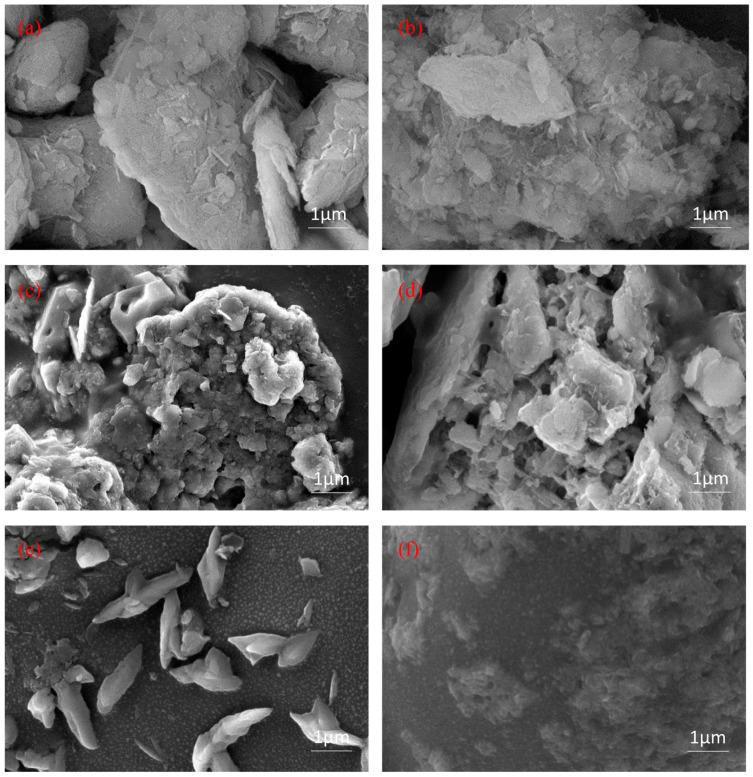
SEM images: (**a**) APT; (**b**) APT modified by 6 mol/L sulfuric acid; (**c**) CSF; (**d**) Starch-lignin-APT; (**e**) lignin-PAM; and (**f**) APT-PAM.

**Figure 4 polymers-15-00265-f004:**
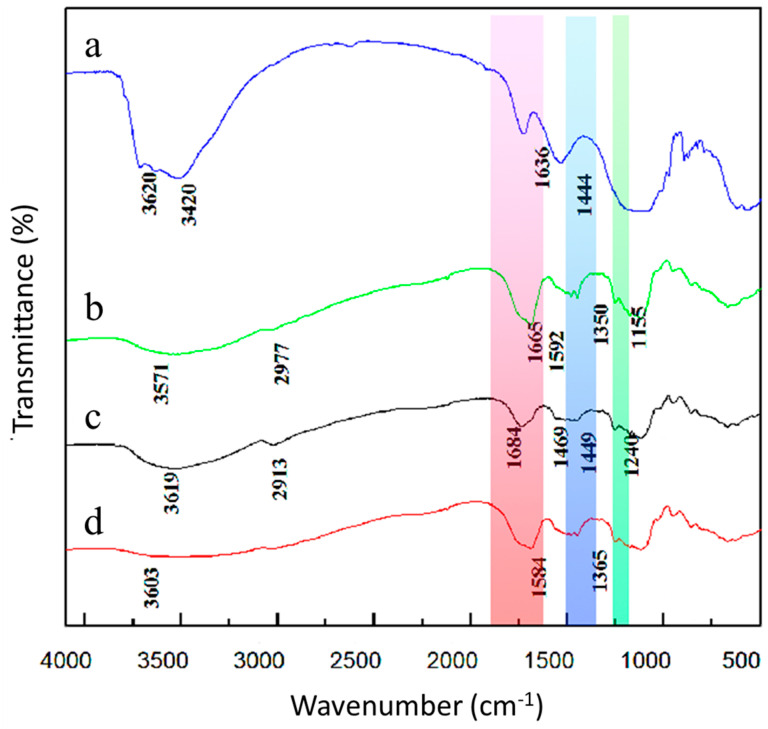
FTIR spectra: (**a**) APT; (**b**) APT−starch−lignin; (**c**) PAM−APT−lignin; and (**d**) PAM−APT−starch−lignin.

**Figure 5 polymers-15-00265-f005:**
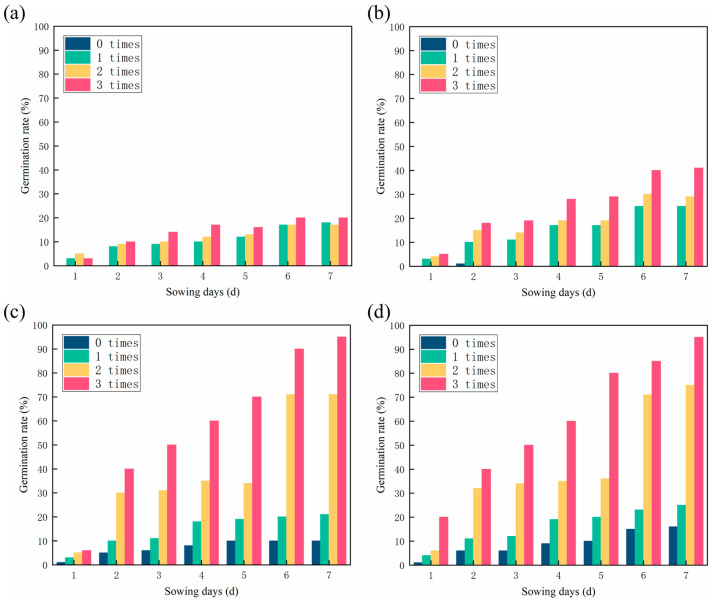
Effects of irrigation times and treatment methods on the germination rate of sandy soil: (**a**) Blank; (**b**) CSF; (**c**) BSF; and (**d**) BAC.

**Figure 6 polymers-15-00265-f006:**
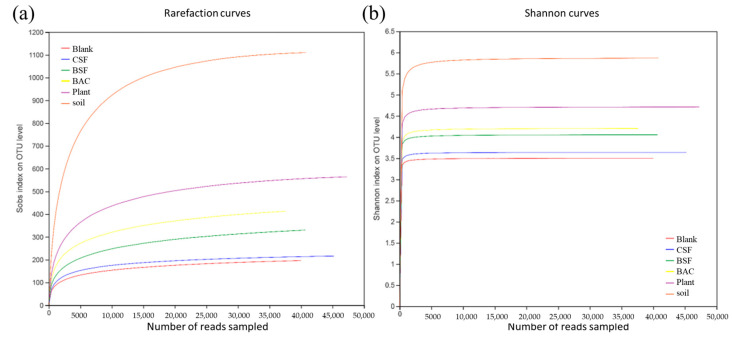
(**a**) Rarefaction curve; and (**b**) Shannon index curve.

**Figure 7 polymers-15-00265-f007:**
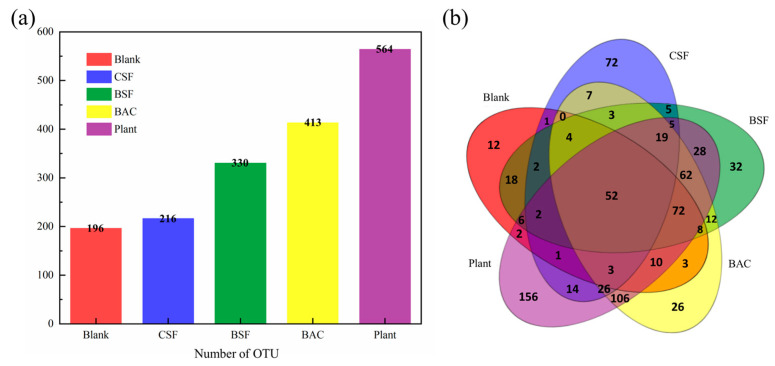
(**a**) Distribution diagram of OTU number; and (**b**) Venn diagram.

**Figure 8 polymers-15-00265-f008:**
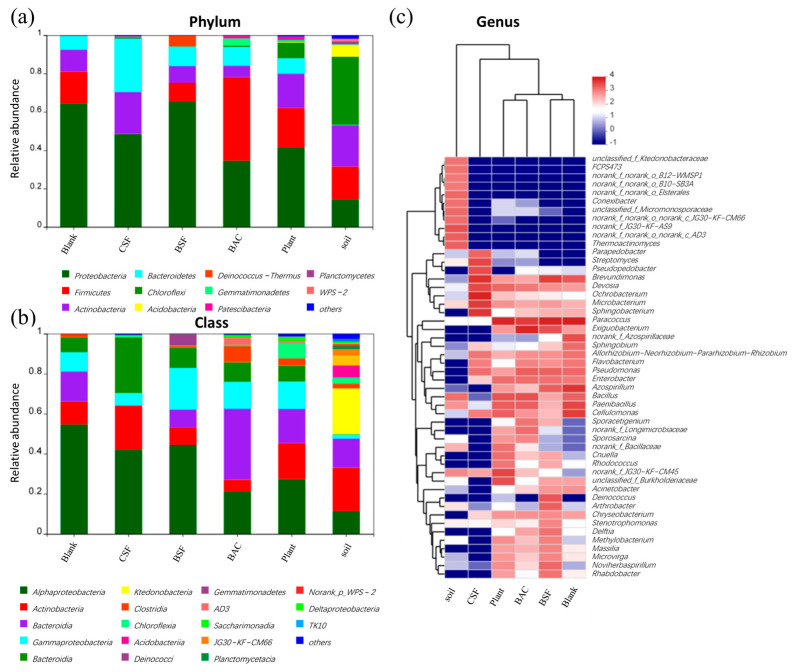
Microbial community composition of each sample based on: (**a**) phylum; (**b**) class; and (**c**) genus level.

**Table 1 polymers-15-00265-t001:** Physicochemical properties of sandy soil.

Properties		Blank	CSF	BSF	BAC
OM (%)		0.06	0.15	20.91	14.09
pH		6.31	7.21	5.32	6.01
TN (‰)		0.02	0.52	1.52	1.42
Olsen-P (mg/kg)		0.81	0.98	25.12	23.38
Avail-K (mg/kg)		1.37	5.13	150.31	122.40
Bulk density (g/cm^−3^)		0.03	1.51	0.61	1.10
Porosity (%)		98.87	43.40	76.98	58.49
Particle size composition (%)	>0.425 mm	0	90	0	40
	0.425~0.25 mm	90	8	20	30
	0.25~0.05 mm	10	2	40	20
	<0.05 mm	0	0	40	10

**Table 2 polymers-15-00265-t002:** Statistics of α diversity index among sandy soil samples.

Sample	Number of OTUs	Coverage (%)	Chao1 Index	ACE Index	Shannon Index
Blank	195	0.99	223.2	218	3.5
CSF	215	0.99	226	230	3.6
BSF	329	0.99	387	374	4.1
BAC	412	0.99	469	463	4.2
Plant	563	0.99	582.2	587	4.7

## Data Availability

Not applicable.

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
