# Peer review of "Preparation and Properties of Bio-Based Attapulgite Copolymer (BAC) Sand-Fixing Material"

_polymers, 2023, doi:10.3390/polym15020265_

Round 1

Reviewer 1 Report

Dear Authors

The presented work is very important and addressed a crucial issue, desertification, one of the world’s most pressing serious environmental problems, poses a serious threat to human survival as well as social, economic, and political development.

Since the environment-friendly sand-fixing method is still a tremendous challenge for preventing desertification. This study developed a bio-based attapulgite copolymer (BAC) by grafting copolymerization of attapulgite, starch, sulfomethyl lignin, and biological mycelia. Water retention, anti-water erosion, and anti-wind erosion tests were conducted to assess the application performance of BAC. Scanning Electron Microscopy (SEM) was then employed to determine the morphology of attapulgite and attapulgite graft copolymer (CSF). The intermolecular interactions in CSF were revealed using Fourier Transform Infrared Spectrum (FT-IR). The role of sand-fixing materials on soil physicochemical properties and seed germination is then discussed based on germination rate experiments. 16S rDNA sequencing technology was used to analyze the differences in microbial communities in each sample group. The results demonstrate that the BAC not only has superior application properties, significantly increaes seed germination (95%) but also promotes soil development by regulating the structure of the soil microbial community. This work can provide novel insights into the design of sand-fixing material for preventing desertification while improving soil feritlity.

The work is applicable and the novelty is enough to attract the attention of the readers. However, the clarity of the work is questionable. 

The main concern is the lack of detailed information about the preparation conditions in the experimental section, which is very confusing and very hard to follow. The authors have to declare separately the preparation conditions of the BAC, CSF, and BSF to ensure the repeatability of the work by other authors. 

In addition, the materials specifications must be provided in the materials section. 

In conclusion, a major revision is required before reconsidering the manuscript for publication. 

Reviewer 2 Report

Corrections:

1.       Please use correct font ( Palatino Linotype) through the manuscript.

2.      Full form of the words like CFU etc needs to be written when initially used in the text ( line 116)

3.      Similar font size should be used in the title for figure 2.

4.      Scale bar in the SEM images should be clear with values on it. (Figure 3)

5.      FTIR peaks should be properly marked denoting the peak value. (figure 4)

6.      In figure 7, graph title is missing for y axis.

7.      In figure 7, the numbers should be denoted in bold.

Detail of correctuon

Check grammer for many sentences the language needs to be clear. (line 16, 58, 60,79, 89, 101, 141, 167,182, 283, 351)

Abstract line 16 recheck this statement

Abstract can be refined.

Line 56-57 Language needs to be refined.

Line 58 Explain in detail those “too many compounds”.

Line 60-61 Repetition of words, needs to be rewritten

Line 68-72 This part needs to be re-read and written for better understanding.

Line 79-80 Sentence needs to be re-written

Line 80-81 'BAC' was appeared first time in the mail text, Please write  Full form needs to be written when mentioned.

Material method should be passive third person past tence.

Line 89 ‘taken from’  Substitute for better language “Procured from”

Line 97 ‘DW’ In the start use complete word then later the short forms.

Line 101-102 Re-write the steps for better understanding.

Line 105 ‘CSF’ Full form needs to be written for the initial text.

Line 108 1×108 should be 1×108

Line 126-127

Check the amount and rewrite for better understanding.

Line 141-142

Check grammar for this sentence.

Line 155-156

This needs to be mentioned in the materials section 2.1 as well.

Line 159-165

Too long sentence, needs to be rewritten. Also with same font size

Line 167-168

Sentence is not clear

Line 172

‘Should be prepared’ Change to “were prepared”

Line 173

Unclear sentence

Line 178

immediately on the sample’Check grammar it should be ” on the collected sample”

Line 182-186

 Too long sentence should be cut tot two short sentences.

Line 282

Select sequences randomly from the sample’ The sentence starts with a verb, please include objective in the sentence.  

Line 284

Please check grammer of ‘leveling off of the rarefaction’

Line 292 ‘OUT’ Full form needs to be written for better understanding.

Line 351 Check grammar and language of the sentence ‘which realized the vision of sand fixation’  

Reviewer 3 Report

The work is good but needs minor revision

1.      Include the equation number.

2.      Add the purity of the used chemicals.

3.      In fig. 3: the SEM images scale bar is not visible. Check it carefully.

4.      In FTIR discussion: There is no discussion of characteristics peaks at 531, 570, 761, and 553 cm-1.

5.      Rewrite the Fig. 6 caption.

6.      In fig. 8: The fig. 8(b) hidden the fig. 8(a) information. So rearrange the figures carefully.

7.      Rewrite the conclusion part.

8.      The authors should check the references carefully. Particularly, references 7, 10, 15, and 16 have no page number. Check it carefully. 

Reviewer 4 Report

polymers-2098605: Preparation and properties of bio-based attapulgite copolymer (BAC) sand-fixing material”.

Research carried out by the authors seems to be important to the development and enhancement of existing information on this subject. Title is consistent with the content of work. Abstract and keywords are prepared in a clear manner and contain the necessary information. The methods used are appropriated. Tables and figures are informative. The work was constructed logically and the study contains a large amount of data. This paper is very interesting and well done. The manuscript figures are of poor quality (it must be corrected). Please also, be sure that all the references cited in the manuscript are also included in the reference list and vice versa. The paper can be accepted for publication after the aforementioned corrections have been made.

Author Response

Some figures in the manuscript have been re-uploaded and some errors have been corrected, thank you again for your review work.

Round 2

Reviewer 1 Report

Dear Authors

The revised version takes into account the raised comments.

I can recommend the revised version for publication. 

Reviewer 2 Report

The manuscript has been improved.